# Research on the Characteristics and Application of Two-Degree-of-Freedom Diagonal Beam Piezoelectric Vibration Energy Harvester

**DOI:** 10.3390/s22186720

**Published:** 2022-09-06

**Authors:** Tianbing Ma, Kaiheng Sun, Shisheng Jia, Fei Du, Zhihao Zhang

**Affiliations:** 1State Key Laboratory of Mining Response and Disaster Prevention and Control in Deep Coal Mines, Anhui University of Science and Technology, Huainan 232001, China; 2College of Mechanical Engineering, Anhui University of Science and Technology, Huainan 232001, China

**Keywords:** piezoelectric, two-degree-of-freedom, energy harvesting, wide-band, auxiliary energy supply

## Abstract

To overcome high periodic maintenance requirements, difficult replacement, and large application limitations of wireless sensor nodes powered by chemical batteries during the vibration control process of stiffened plates, a two-degree-of-freedom diagonal beam piezoelectric vibration energy harvester was proposed. Multidimensional energy harvesting and broadband work are integrated into one structure through the combined action of oblique angle, mass blocks, and piezoelectric beam. The mechanical model of the beam is established for theoretical analysis; the output characteristics of the structure are analyzed by finite element simulation; a piezoelectric energy harvesting experimental bench is built. The results show that: The structure has a wider harvesting band, multi-order resonant frequency, multi-dimensional energy harvesting, and higher output voltage and power than the traditional cantilever structures. The output performance of the specimens with 45° oblique angle, 5 g:5 g mass ratio, and 0.2 mm thickness of piezoelectric substrate is good in the frequency band of 10~40 Hz. When the excitation frequency is 28 Hz, the output voltage of the sextuple array structure reaches 19.20 V and the output power reaches 7.37 mW. The field experiments show that the harvester array can meet the requirements of providing auxiliary energy for wireless sensor nodes in the process of active vibration control of stiffened plates.

## 1. Introduction

Over the years, stiffened plate structures have been widely used in many industrial fields such as aerospace, ships, bridges, and automobiles due to their high mechanical strength, durability, and economy [1,2]. When the stiffened plate structure is disturbed by external forces, it is easy to resonate with the external environment, resulting in damage to the structure itself, thus reducing its working life, and radiating noise around, so it is necessary to actively control the vibration of the stiffened plate [3,4].

At present, most of the wireless sensor elements used in active control are powered by traditional chemical batteries. This power supply mode has irreversible defects, its service life is short and needs to be replaced regularly, so it is difficult to apply to sealed places that are not easy to disassemble [5]. In addition, the wireless sensor elements for the wireless control process of stiffened plate vibration are widely used, and the replacement of batteries requires a lot of manpower and material resources, which limits the application of active control [6].

In order to solve the problem of power supply in wireless data transmission of active vibration control of stiffened plate, combined with vibration energy harvest technology, considering that the vibration frequency band of commonly used stiffened plate is in the range of 5~40 Hz, 80~120 Hz, and 260~300 Hz, it is proposed to transform and utilize vibration energy to realize vibration absorption and assist power supply for wireless transmission components [7,8].

According to different conversion mechanisms, the current vibration energy harvesting technology can be divided into three types: piezoelectric, electrostatic, and electromagnetic [9,10,11]. Among them, the piezoelectric vibration energy harvester has been widely concerned and applied because of its simple structure, long service life, no electromagnetic interference, high energy density, and being able to absorb vibration to a certain extent [12,13,14,15].

Rui [16] proposed a self-tuning rotating piezoelectric energy harvest structure, which achieves the function of self-tuning by changing the centrifugal force to achieve wide-band harvesting. Lu [17] proposed a rotating piezoelectric vibration energy harvest structure with an elastic structure, which uses the cooperation of structures to produce nonlinear vibration to adjust the resonant frequency and realize the wide-band harvesting. Wang [18] proposed a piezoelectric cantilever structure with an elastic amplifier. By using the wave deformation of the linear elastic element, the influence of vibration on the piezoelectric chip is magnified in a physical way and realizes the wide-band energy harvesting. Ramírez [19] proposed a piezoelectric vibration energy harvest structure of ultra-low frequency tuned composite cantilever beam, which can adjust the mass through the interaction of multiple cantilever beams to achieve wide-band energy harvesting. Gulec [20] proposed a spiral blade array piezoelectric energy harvest structure, which adjusts the vibration frequency by changing the parameters of the structure, so that the vibration frequency of the whole device changes and realizes the wide-band energy harvesting. Yang [21] proposed a kind of annular piezoelectric disk vibration energy harvest structure, which can make full use of the utilization of piezoelectric materials, improve the output power of the energy harvest structure, and further improve the harvesting performance.

Hou [22] proposed a multi-directional piezoelectric vibration energy harvest device, which adopts a rainbow piezoelectric structure to harvest vibration energy in different directions. Wang [23] proposed a multi-dimensional piezoelectric energy harvesting structure of a composite beam with a spring, which can harvest multi-dimensional vibration energy through the interaction between the magnetic mass at the end of the main beam and the mass on the auxiliary beam. Deng [24] proposed a kind of piezoelectric vibration energy harvest structure of a multi-mode double-branch composite beam, which realizes multi-dimensional vibration energy harvesting through the joint action of two branch beams and the main beam. Nguyen [25] designed a cylindrical piezoelectric vibration energy harvester, which uses the deformation of the cylindrical surface under the action of multi-dimensional vibration to deform the piezoelectric chip, so as to realize the harvesting of multi-dimensional vibration energy. Li [26] proposed a hybrid cantilever piezoelectric energy harvest structure with multi-mode bonded MFC, by determining one beam as the main beam, changing the number of other beams and the mass of the end mass, the multi-dimensional energy harvesting is realized. Sofiane [27] designed a dual-frequency vibration-based energy harvester with a collection frequency band of 50–100 Hz for ultra-low power devices in industrial environments, which can resonate at dual frequencies of 63.3 and 76.4 Hz, equipped with a power management module, the harvester can provide 500 µW power, and it can output 253 μW of power in a white noise environment of 50–100 Hz with a power density of 0.005 g^2^/Hz. Andrius [28] proposed an energy harvester based on a piezoelectric sawtooth cantilever array, by placing a Z-shaped mass block in the middle of the Z-shaped beam, it can collect bidirectional vibration energy, and there are four resonant frequencies within 10–160 Hz, and it was confirmed that the maximum output power of its array structure can reach 8.06 mW. Pranjal [29] designed a piezoelectric vibration energy harvester array for vehicle suspension and made a prototype, which increases the output voltage of the harvester by arraying 20 piezoelectric chips, and can be installed on any vehicle suspension system. Mani [30] proposed a MEMS energy harvester for mechanical and contactless magnetic excitation with output power up to 74.11 μW under gyromagnetic excitation below 50 Hz, and its output power at resonance can reach 139.39 μW.

At present, the optimization of piezoelectric energy harvest devices is mainly from three aspects: harvesting frequency band, harvesting dimension, and harvesting efficiency [31], but most of the piezoelectric energy harvest devices are only optimized in a single aspect. Considering that the vibration in the application of piezoelectric energy harvest devices is complex and changeable, a two-degree-of-freedom diagonal beam piezoelectric vibration energy harvest structure is proposed. Compared with the commonly used rectangular and trapezoidal cantilever structure, this structure can not only broaden the harvesting frequency band but also increase the dimension of energy harvesting, realize multi-dimensional vibration energy harvesting, and improve the output power.

The energy harvesting structure is mainly composed of a piezoelectric beam base, two piezoelectric slices, and two mass blocks. There is a certain angle between the elastic matrix and the horizontal surface, the piezoelectric material is pasted on the elastic matrix, and the mass is placed in the middle and end of the elastic matrix. The energy harvesting structure can not only effectively reduce the resonant frequency, but also approach the first two resonant frequencies of the structure, thus broadening the harvesting frequency band of the energy harvester. By setting the intermediate mass, the strain distribution on the piezoelectric beam is more uniform and the utilization rate of the piezoelectric material is improved. By tilting the elastic matrix at a certain angle, the response ability of the structure to vibration is effectively improved, and multi-directional vibration energy harvesting is realized.

In this paper, the two-degree-of-freedom diagonal beam piezoelectric vibration energy harvester is studied from three levels: theory, simulation, and experiment. The mechanical model of the structure is established, and the main factors affecting the power generation performance and operating frequency bandwidth of the structure are analyzed and obtained; through the finite element simulation method, the structure is compared with the two commonly used energy harvest structures, the superiority of the two-degree-of-freedom diagonal beam piezoelectric vibration energy harvest structure is verified, the effects of load resistance, vibration excitation intensity and size parameters on the output performance of the structure are analyzed, and the size of the structure is preliminarily determined; the test bench was built, the samples of the structure were made, the actual output performance of the structure was tested, the experiment verified the theoretical and simulation results, and the array and application experiments of the structure were carried out, and it is proved that the harvester can meet the requirements of providing auxiliary energy for wireless sensor nodes in the process of active vibration control of stiffened plates.

## 2. Modeling and Theoretical Analysis

The structural diagram of the two-degree-of-freedom diagonal beam piezoelectric vibration energy harvester is shown in Figure 1. The structure consists of a fixed clamping end, two piezoelectric beams, and two mass blocks connected obliquely; make the piezo beam at an angle to the fixed clamping end, mass 1 is placed obliquely between piezo beams 1 and 2, in the same direction as the piezo beam, and mass 2 is placed vertically at the end of piezo beam 2. The length of piezo beams 1 and 2 is *b* and *a*, respectively, the length of the whole beam structure is *L*, the width is *c*, and the equivalent masses of mass 1 and mass 2 are *m*_1_ and *m*_2_, with *m*_1_:*m*_2_ = *θ*. Take a section at any position *x* on the piezo beam 1, and the schematic diagram of its cross-section is shown in Figure 2. The thickness of the piezoelectric beam base is *h*_1_, and the thickness of the piezo is *h*_2_.

### 2.1. Generation Performance Analysis

In order to analyze the power generation performance of the structure, a simplified mathematical model is established. Establishing the static mathematical model of piezo beam 1. Considering that the angle has little influence on the structural performance in the static analysis, the structure is equivalent to a classical cantilever beam for simplified analysis. Assuming that the external force on the piezo beam is equivalent to the resultant force *F* applied to the free end of the structure, the bending moment on the piezoelectric beam at any position *x* on the beam is [32,33]:(1)M(x)=F(L−x)

The static deflection equation at section *x* is:(2)$$z\left(x\right)=\iint \frac{M\left(x\right)}{EI}dx$$
where *E* is the elastic modulus, *I* is the moment of inertia of the selected section to the y-axis, and *EI* is the bending stiffness.

Piezo beam 1 is forced to bend and deform by external force, and its deformation neutral layer position is:(3)$$z_{0}=\frac{E_{1}h_{1}\left(h_{2}+\frac{h_{1}}{2}\right)+E_{2}\frac{h_{2}^{2}}{2}}{E_{1}h_{1}+E_{2}h_{2}}$$
where *E*_1_ is the elastic modulus of the piezoelectric beam base, and *E*_2_ is the elastic modulus of the piezoelectric material.

Assuming that the equivalent bending stiffness of the piezo beam is *E*_12_*I*_12_, then:(4)$$E_{12}=E_{1}\int_{0}^{c}\int_{h_{1}}^{z_{0}}zdzdy+E_{2}\int_{0}^{c}\int_{z_{0}}^{h_{2}}zdzdy$$
(5)$$I_{12}=\int_{0}^{c}\int_{-h_{1}}^{h_{2}}zdzdy$$
where *E*_12_ is the equivalent elastic modulus of the piezoelectric beam, and *I*_12_ is the polar moment of inertia of the piezo beam section about the Y axis.

The static deflection equation of piezo beam 1 at section *x* is:(6)$$z_{12}\left(x\right)=\iint \frac{M\left(x\right)}{E_{12}I_{12}}dx$$

At time *t*, the curvature equation at any *x* of the beam is:(7)$$\frac{1}{r_{12}\left(x,\;t\right)}=\frac{\partial^{2}z_{12}\left(x,\;t\right)}{\partial x^{2}}$$
where *r*_12_(*x*, *t*) is the radius of curvature at position *x* at time *t* of the beam, and *z*_12_(*x*, *t*) is the dynamic deflection equation of the beam.

Considering that the measured voltage is the output under the resonant frequency of the piezoelectric plates, the generation performance of the piezo beam in resonance state is analyzed, and the analysis of its vibration mode function shows that:(8)$$\frac{1}{r_{12}\left(x\right)}=\frac{d^{2}z_{12}\left(x\right)}{dx^{2}}$$
where *z*_12_(*x*) is the vibration mode function of the beam, which can be calculated by the static deflection equation of the beam.

Considering the relationship between stress and strain in the piezo sheet, it can be concluded that:(9)$$T_{1}=E_{2}\left(S_{1}-g_{31}D_{3}\right)$$
(10)$$e_{3}=\beta_{33}^{T}D_{3}-g_{31}T_{1}$$

Among them:(11)$$S_{1}=-rz,\;\beta_{33}^{T}=\frac{1}{\varepsilon_{33}^{T}}$$
where *T*_1_ is the stress on the piezo along the *x*-axis, *S*_1_ is the strain, *E*_2_ is the elastic modulus of piezoelectric material, *g*_31_ is the piezoelectric constant, *D*_3_ is the potential shift along the z-axis, *e*_3_ is the electric field strength, *r* is the radius of curvature at the *x* position of the beam, $\varepsilon_{33}^{T}$ is the dielectric constant, and $\beta_{33}^{T}$ is the dielectric isolation rate.

Through the above formula, the electric field distribution equation can be obtained:(12)$$e_{3}=\beta_{33}^{T}D_{3}-g_{31}E_{p}\left(r_{12}\left(x\right)z+g_{31}D_{3}\right)$$

In addition, the voltage equation is:(13)$$V=\int_{-h_{2}}^{z_{0}}e_{3}dz=-\frac{1}{2}g_{31}E_{2}\left(z_{0}^{2}-h_{2}^{2}\right)r_{12}\left(x\right)+\left(z_{0}+h_{2}\right)\left(g_{31}^{2}D_{3}E_{2}+\beta_{33}^{T}D_{3}\right)$$

It can be concluded that the charge equation is:(14)$$Q_{12}=\int_{0}^{c}\int_{0}^{1}D_{3}dzdy=CV$$

The theoretical output voltage can be obtained by combining the above formula:(15)$$V=\frac{cb-2\left(z_{0}+h_{2}\right)\left(g_{31}^{2}E_{2}+\beta_{33}^{T}\right)C}{g_{31}E_{p}\left(z_{0}^{2}-h_{2}^{2}\right)r_{12}\left(x\right)}$$
where *C* is the equivalent capacitance of piezoelectric material

From the above derivation, it can be seen that the theoretical output voltage of the structure is inversely proportional to the radius of curvature. The radius of curvature represents the bending degree of the piezo beam. The greater the bending degree, the smaller the radius of curvature, and the greater the theoretical output voltage. That is, when the natural frequency of the piezoelectric beam approaches or overlaps with the external excitation frequency, the deformation of the beam is the largest, and the theoretical output voltage is the largest.

### 2.2. Broadband Characteristic Analysis

Through reasonable structural parameter design, the first-order and second-order natural frequencies of the harvester are close to each other during vibration, so as to achieve the purpose of low-frequency and broadband energy harvesting of the piezoelectric energy harvester. In order to simplify the analysis of the frequency band characteristics of the two-degree-of-freedom diagonal beam piezoelectric vibration energy harvester, considering that the angle has little effect on the frequency characteristics of the structure, it is ignored in the derivation process. As shown in Figure 1, it is assumed that the equivalent masses of mass 1 and 2 are *m*_1_ and *m*_2_, respectively; the equivalent stiffness of piezoelectric beams 1 and 2 is *k*_1_ and *k*_2_.

When analyzing the frequency characteristics, the two-degree-of-freedom diagonal beam can be simplified as a spring-mass damping two-degree-of-freedom system under free vibration. The vibration equation of the system is [34]:(16)$$\left\{ \begin{array}{l} m_{1}{\ddot{u}}_{1}+2k_{1}u_{1}-k_{1}u_{2}=0\\ m_{2}{\ddot{u}}_{2}-k_{2}u_{1}+k_{2}u_{2}=0 \end{array}\right.$$

The equivalent stiffness of the piezoelectric beam and the equivalent bending stiffness of its section are related as follows:(17)$$\left\{ \begin{array}{l} k_{1}=\frac{3E_{12}I_{12}}{b^{3}}\\ k_{2}=\frac{3E_{12}^{\prime }I_{12}^{\prime }}{a^{3}} \end{array}\right.$$
where $E_{12}I_{12}$ and $E_{12}^{\prime }I_{12}^{\prime }$ are the equivalent bending stiffness of piezo beams 1 and 2. Let:(18)$$M=\left[\begin{array}{cc} m_{1} & 0\\ 0 & m_{2} \end{array}\right];K=\left[\begin{array}{cc} 2k_{1} & -k_{1}\\ -k_{2} & k_{2} \end{array}\right];U=\left[\begin{array}{c} u_{1}\\ u_{2} \end{array}\right]$$

The vibration equation of the system can be reduced to:(19)$$M\ddot{U}+KU=0$$

According to the knowledge of linear algebra, the determinant of the matrix on the left of the equal sign is:(20)$$\left|\begin{array}{cc} K_{11}-M_{11}w^{2} & K_{12}\\ K_{21} & K_{22}-M_{22}w^{2} \end{array}\right|=0$$

The frequency characteristic equation of the system can be obtained as follows:(21)$$m_{1}m_{2}w^{4}+\left(2m_{2}k_{1}+m_{1}k_{2}\right)w^{2}+\left(2k_{1}k_{2}-k_{1}^{2}\right)=0$$

The eigenvalue of the equation can be solved; that is, the first two order system frequencies of the system are:(22)$$w_{1}^{2}=\frac{\left(2m_{2}k_{1}+m_{1}k_{2}\right)-\sqrt{{\left(2m_{2}k_{1}+m_{1}k_{2}\right)}^{2}-4m_{1}m_{2}k_{1}\left(2k_{2}-k_{1}\right)}}{2m_{1}m_{2}}$$
(23)$$w_{2}^{2}=\frac{\left(2m_{2}k_{1}+m_{1}k_{2}\right)+\sqrt{{\left(2m_{2}k_{1}+m_{1}k_{2}\right)}^{2}-4m_{1}m_{2}k_{1}\left(2k_{2}-k_{1}\right)}}{2m_{1}m_{2}}$$

Then, the relationship between the frequencies of the first two orders of the system can be obtained as follows:(24)$$\frac{w_{1}^{2}}{w_{2}^{2}}=\frac{\left(2m_{2}k_{1}+m_{1}k_{2}\right)-\sqrt{{\left(2m_{2}k_{1}+m_{1}k_{2}\right)}^{2}-4m_{1}m_{2}k_{1}\left(2k_{2}-k_{1}\right)}}{\left(2m_{2}k_{1}+m_{1}k_{2}\right)+\sqrt{{\left(2m_{2}k_{1}+m_{1}k_{2}\right)}^{2}-4m_{1}m_{2}k_{1}\left(2k_{2}-k_{1}\right)}}$$

From the above derivation, it can be seen that the first two frequencies of the diagonal beam with two degrees of freedom used in this paper are related to the size parameters of the structure and the mass of the two mass blocks. Therefore, through reasonable design of various structural parameters of the device, the first- and second-order vibration frequencies are close to each other, so as to broaden the frequency band and achieve high-efficiency energy harvesting.

### 2.3. Amplitude Response Characteristic Analysis

On the basis of the above analysis, assuming that mass 2 is subjected to a variable force *F*_0_ sin *wt* from the vertical direction, the two-degree-of-freedom diagonal beam piezoelectric vibration energy harvester can be regarded as an undamped two-degree-of-freedom forced vibration system [35,36].

Through numerical analysis of the system, the vibration equation of the two-degree-of-freedom beam system can be obtained as follows:(25)$$\left[\begin{array}{cc} m_{1} & 0\\ 0 & m_{2} \end{array}\right]\left[\begin{array}{c} {\ddot{u}}_{1}\\ {\ddot{u}}_{2} \end{array}\right]+\left[\begin{array}{cc} 2k_{1} & -k_{1}\\ -k_{2} & k_{2} \end{array}\right]\left[\begin{array}{c} u_{1}\\ u_{2} \end{array}\right]=\left[\begin{array}{c} 0\\ F_{0}\sin wt \end{array}\right]$$

The equation is a system of second-order constant coefficient inhomogeneous linear differential equations. It can be assumed that its particular solution is:(26)$$\left\{ \begin{array}{l} u_{1}=B_{1}\sin wt\\ u_{2}=B_{2}\sin wt \end{array}\right.$$
where *B*_1_ and *B*_2_ are the amplitudes of piezo beams 1 and 2. Taking the second derivative of the above equation with respect to time, the system acceleration can be obtained as:(27)$$\left\{ \begin{array}{l} {\ddot{u}}_{1}=-B_{1}w^{2}\sin wt\\ {\ddot{u}}_{2}=-B_{2}w^{2}\sin wt \end{array}\right.$$

By substituting Equations (26) and (27) into Equation (25), we can obtain:(28)$$B_{1}=\frac{k_{1}}{\left(2k_{1}-w^{2}m_{1}\right)\left(k_{2}-w^{2}m_{2}\right)-k_{1}k_{2}}$$
(29)$$B_{2}=\frac{2k_{1}-w^{2}m_{1}}{\left(2k_{1}-w^{2}m_{1}\right)\left(k_{2}-w^{2}m_{2}\right)-k_{1}k_{2}}$$

Combined with Section 2.2, then:(30)$${w_{1}}^{2}+{w_{2}}^{2}=\frac{k_{2}}{m_{2}}+\frac{2k_{1}}{m_{1}}$$
(31)$${\left({w_{1}}^{2}-{w_{2}}^{2}\right)}^{2}=\frac{k_{2}^{2}}{m_{2}^{2}}+\frac{4k_{1}^{2}}{m_{1}^{2}}$$

By substituting Equations (30) and (31) into (28) and (29), respectively, it can be obtained that the amplitude response functions of the amplitudes of piezo beams 1 and 2 with respect to the excitation frequency are:(32)$$B_{1}\left(w\right)=\frac{\left(w_{1}^{2}+w_{2}^{2}\right)+\sqrt{{\left(w_{1}^{2}+w_{2}^{2}\right)}^{2}-8w_{1}^{2}w_{2}^{2}}}{4m_{2}\left(w^{2}-w_{1}^{2}\right)\left(w^{2}-w_{2}^{2}\right)}$$
(33)$$B_{2}\left(w\right)=\frac{\left(w_{1}^{2}+w_{2}^{2}\right)-2w^{2}+\sqrt{{\left(w_{1}^{2}+w_{2}^{2}\right)}^{2}-8w_{1}^{2}w_{2}^{2}}}{2m_{2}\left(w^{2}-w_{1}^{2}\right)\left(w^{2}-w_{2}^{2}\right)}$$

After the numerical simulation of *B*_1_(*w*) and *B*_2_(*w*), the results can be obtained: In the amplitude response analysis of the first mode of the two-degree-of-freedom diagonal beam piezoelectric vibration energy harvester, the amplitude at the end of piezoelectric beam 2 is obviously larger than that at the end of the piezoelectric beam 1, and the fatigue damage of the piezoelectric material is also greater. In the structural performance exploration, under the premise of the same total mass, the mass of mass block 2 should be appropriately reduced to increase the service life of the energy harvester.

## 3. Finite Element Simulation Analysis of Structural Characteristics

In order to explore the output performance of the two-degree-of-freedom diagonal beam piezoelectric vibration energy harvester, the finite element simulation analysis of the structure is carried out by COMSOL Multiphysics software [37]. The performance parameters of corresponding materials are shown in Table 1.

### 3.1. Horizontal Comparison of Different Structures

The models of the rectangular cantilever beam, variable cross-section cantilever beam, and two-degree-of-freedom diagonal beam with the same volume of the piezo beam and piezoelectric beam base are established for simulation analysis. Select the fixed end face parallel to the Z axis, and apply the body load (excitation strength) along the Z axis. To ensure the reliability of the results, set the same excitation intensity and load resistance.

As shown in Figure 3, the output voltage of the diagonal beam with two degrees of freedom is 18.27 V at the excitation frequency of 6 Hz and 41.48 V at 40 Hz. Compared with the rectangular cantilever beam and variable cross-section cantilever beam, due to the mutual tuning effect of mass 1 and 2, the first- and second-order resonant frequencies of the two-degree-of-freedom diagonal beam are closer, which greatly widens the structure harvesting frequency band. At the same time, the beam is in a state of two degrees of freedom when it is vibrated, and the front and rear ends of the piezoelectric beam have large bending deformation, which greatly improves the utilization of piezoelectric materials and makes the output voltage of the structure higher.

Considering that the natural frequencies of vibration in practical applications are complex and changeable, and are distributed continuously in multiple bands, the harvesting frequency band of the vibration energy harvesting device is required to be relatively wide, at the same time, compared with the power consumption of wireless sensor elements, the device is required to have higher output performance. After horizontal comparison with the rectangular cantilever beam and variable cross-section cantilever beam, it is concluded that the harvesting performance of the two-degree-of-freedom diagonal beam is better, which is more suitable for supplying energy for current wireless sensor elements.

### 3.2. Correlation Analysis of External Factors

This section explores the impact of external factors on structural harvesting performance. The main external factors that affect the experimental output performance are load resistance and excitation intensity.

#### 3.2.1. Load Resistance

Using the method of control variables, keep the other conditions constant, set different load resistance values, carry out simulation analysis, and explore the output voltage of different load corresponding structures when the excitation frequency is 40 Hz. The result is shown in Figure 4.

It can be seen from Figure 4 that as the load resistance increases, the output voltage of the structure increases because when the alternating voltage is generated by the alternating deformation of the piezo beam during the vibration process, the impedance is generated simultaneously within the piezo beam. The external load resistance in the circuit and the internal resistance of the piezo beam divide the voltage produced by the piezoelectric effect. As the external load resistance increases, the external voltage dividing capability increases, and the output voltage of the structure increases. With the increase in load resistance, the increasing rate of output voltage decreases gradually, that is, there is optimal output power. Assuming that the output power of the structure is *P_R_*, then:(34)$$P_{R}=\frac{V_{R}^{2}}{R}$$
where *V_R_* is the external load voltage and *R* is the load resistance. Assuming that the internal impedance of the piezo beam is *R_r_*, then:(35)$$V_{R}=\frac{VR}{R+R_{r}}=V-\frac{VR_{r}}{R+R_{r}}$$

Combining Equations (34) and (35), we can get:(36)$$P_{R}=\frac{V^{2}}{R+R_{r}}-\frac{V^{2}R_{r}}{(R+R_{r}{)}^{2}}$$

It can be seen that the relationship between the power and the reciprocal of resistance is a quadratic function, so there is an optimal load resistance to maximize the output power. According to the requirements of the later article, the simulation and experimental results should be obvious, when the load resistance *R* is 50 kΩ, both the output voltage and output power of the structure have good values, so the best external load of the energy harvester is 50 kΩ.

#### 3.2.2. Incentive Intensity

The exploration of vibration excitation intensity is similar to the exploration of the correlation of load resistance. Take the same excitation frequency and external load resistance (50 kΩ), set the auxiliary scanning, take the vibration excitation intensity as 0.2 to 2 g (step size as 0.2 g), and conduct a simulation. The results are shown in Figure 5.

It can be seen from the figure that the output voltage increases linearly with the increase in vibration excitation intensity. The main reason is that the increase in excitation intensity increases the amplitude of the structure, so that the end displacement of the structure increases linearly in the steady state, and the tension-compression bending deformation of the piezoelectric material increases, so that the output voltage increases linearly. Since the resistance of the structure is constant, as the output voltage increases, its output power increases, and the power is a quadratic function of the voltage, the excitation intensity is positively correlated with the output voltage and power.

Based on the above conclusion, when selecting the excitation strength, after meeting a certain output performance, the physical properties of the piezoelectric beam need to be considered. Combined with the equivalent tensile strength of the piezoelectric beam, the vibration excitation strength is 1 g in the simulation process. In the experiment, the piezo beam is made of piezo and piezoelectric beam base bonded with glue, and its physical properties are quite different from those in the simulation. Therefore, the excitation strength needs to be further explored and determined in the experiment.

### 3.3. Structural Design of Beam

Through the above mechanical modeling and frequency characteristic analysis of the beam, it is known that the first two resonant frequency of the structure is closely related to the geometric parameters of the structure. Through the optimization of its dimension parameters, the first- and second-order resonant frequencies of the structure can be close to each other, and the harvesting frequency band of the energy harvester can be broadened. Therefore, it is necessary to select and optimize the geometric dimensions of the structure.

#### 3.3.1. Structural Dimension Design

Considering the coupling effect of each parameter on the results, the optimal size of the structure cannot be obtained directly from the numerical modulus but must be obtained through parameter optimization considering the influence of each size. Therefore, the length *b* and *a*, the width *c*_1_ and *c*_2_, the thickness *h*_1_ and *h*_2_, the equivalent mass *m*_1_ and *m*_2_, and the oblique angle *α* are taken as the design variables, respectively, and the minimum ratio of the first-order resonance frequency of the energy harvester is taken as the goal, create the objective function [38,39,40,41]:(37)$$\eta_{\omega }=\min f\left(h_{1},b,c_{1},j,c_{2},h_{2},m_{1},m_{2},\alpha \right)$$

Among them:(38)$$f\left(h_{1},b,c_{1},j,c_{2},h_{2},m_{1},m_{2},\alpha \right)=\frac{\sqrt{\left(2m_{2}k_{1}+m_{1}k_{2}\right)-\sqrt{{\left(2m_{2}k_{1}+m_{1}k_{2}\right)}^{2}-4m_{1}m_{2}k_{1}\left(2k_{2}-k_{1}\right)}}}{\sqrt{\left(2m_{2}k_{1}+m_{1}k_{2}\right)+\sqrt{{\left(2m_{2}k_{1}+m_{1}k_{2}\right)}^{2}-4m_{1}m_{2}k_{1}\left(2k_{2}-k_{1}\right)}}}$$

Analyze the objective function, consider that the oblique angle has the same effect on the rest of the geometric dimensions, simplify the calculation, and ignore the influence of the oblique angle, then:(39)$$k_{1}=\frac{c_{1}\left[E_{1}h_{1}^{3}+E_{2}\left(h_{2}^{3}+3h_{1}^{2}h_{2}+3h_{1}h_{2}^{2}\right)\right]}{b^{3}}$$
(40)$$k_{2}=\frac{c_{2}\left[E_{1}h_{1}^{3}+E_{2}\left(h_{2}^{3}+3h_{1}^{2}h_{2}+3h_{1}h_{2}^{2}\right)\right]}{j^{3}}$$

Combined with the research background, the spatial structure requirements of the energy harvester and the strength limit of the piezoelectric material, the following constraints can be set according to the maximum allowable tensile stress and tensile strength of the piezo during bending deformation:$$\begin{array}{c} 0.2\;\mathrm{mm}\leq h_{1}\leq 0.5\;\mathrm{mm}\\ 20\;\mathrm{mm}\leq b\leq 50\;\mathrm{mm}\\ 20\;\mathrm{mm}\leq c_{1}\leq 50\;\mathrm{mm}\\ 20\;\mathrm{mm}\leq a\leq 50\;\mathrm{mm}\\ 5\;\mathrm{mm}\leq c_{2}\leq 20\;\mathrm{mm}\\ 0.2\;\mathrm{mm}\leq h_{2}\leq 0.3\;\mathrm{mm}\\ 1\;g\leq m_{1}\leq 15\;g\\ 1\;g\leq m_{1}\leq 15\;g\\ 30{}^{\circ}\leq \alpha \leq 60{}^{\circ} \end{array}$$

The sequential quadratic programming method is used to optimize the objective function. The initial parameters before optimization are as follows: the thickness of the piezoelectric beam base is 0.2 mm; the length of piezo 1 is 38 mm, the width is 12 mm and the thickness is 0.2 mm; the size parameters of piezo 2 are the same as piezo 1, and the equivalent mass of mass 1 and 2 is 5 g.

After optimization, the size parameters are as follows: the length of piezo 1 is 38 mm, the width is 12 mm, the thickness is 0.2 mm; the length of piezo 2 is 38 mm, the width is 12.2 mm, and the thickness is 0.2 mm. The thickness of the piezoelectric beam base, oblique angle, and the mass of mass 1 and 2 still need to be confirmed by further study.

The structural performance is not obviously related to the shape of the two mass blocks, but to their mass. Set the length and width size of the mass block to 20 mm × 10 mm, and change the mass of the mass 1 and 2 by changing their thickness *h*_3_ and *h*_4_, respectively; for the piezoelectric beam base, combined with the size of piezo 1, 2 and mass 1, on the basis of not affecting the structural performance, reserve a certain bonding space, and set it to 100 mm × 15 mm; the structural performance is not related to the geometric dimensions of the fixed clamping end and the fixed end of the end mass, and their geometric dimensions are set to 15 mm × 15 mm × *h*_1_. The structure diagram of the two-degree-of-freedom diagonal beam drawn based on the above data is shown in Figure 6.

#### 3.3.2. The Oblique Angle of the Beam

Ensure that the shape and material of the piezoelectric beam base and the piezo sheets remain unchanged, and change the oblique angle of the structure to explore its influence on the harvesting performance of the structure. In combination with the constraints given in the previous section and the conclusions in Section 3.2, take three groups of test pieces with oblique angles of 30°, 45°, and 60°, apply 1 g excitation strength and 50 kΩ external resistance load, respectively, and obtain the harvest performance analysis results, as shown in Figure 7.

It can be seen from the figure that when the oblique angle of the structure is 45°, the output performance is the best, the first-order resonance frequency is 6 Hz, the corresponding output voltage is 18.27 V, the second-order resonance frequency is 40 Hz, and the corresponding output voltage is 41.48 V. That is, when the oblique angle of the beam is 45°, the mutual coupling effect of mass blocks 1 and 2 on the piezoelectric beam reaches the optimum, the tensile and compressive stress on the piezoelectric material reaches the maximum, and the output voltage of the structure reaches the maximum.

In addition, it can be seen from Figure 7 that the resonant frequency of the structure has little correlation with the oblique angle, which is consistent with the assumption that the angle has little effect on the frequency band in the previous chapter. Therefore, the specimen with an oblique angle of 45° was selected for subsequent simulation analysis. Considering that the simulation is unreliable, the conclusion is verified by experiments in the following sections.

#### 3.3.3. Mass Ratio of Mass 1 and 2

A structural specimen with an oblique angle of 45° was taken to explore the changes in the harvesting performance of the two-degree-of-freedom diagonal beam structure under the action of different mass ratios of the mass blocks. Take the definition *θ* = *m*_1_:*m*_2_ of the mass-block ratio in Chapter 2, keep the other parameters consistent, set up multiple groups of structures with different mass ratios, and obtain the analysis results of the harvesting performance as shown in Figure 8.

The output voltage data of the structures with different mass ratios at the resonance frequency are shown in Table 2.

It can be obtained from the above: within the acceptable range of piezoelectric materials, changing the mass of the mass block can effectively adjust the resonant frequency of the piezoelectric energy harvesting structure. The greater the mass of the mass block, the lower the resonance frequency of the structure and the better the harvesting performance.

To sum up, for the two-degree-of-freedom diagonal beam structure, the main factor affecting its resonant frequency is the total mass of the mass block. The output performance of the structure is best when the masses are of equal mass. In order to optimize the output characteristics of the piezoelectric energy harvester and realize the broadband harvesting function, considering the maximum tensile stress and service life that the piezoelectric material can withstand, *θ* is set to 5 g:5 g. Considering that the data come from a finite element simulation and have a certain deviation from the actual situation, the conclusion is verified by experiments in the following paper.

#### 3.3.4. Thickness of Piezoelectric Beam Base

The thickness of the piezoelectric beam base has a strong influence on the stiffness and strength of the structure. Keep other parameters of the structure unchanged, change the thickness *h*_1_ of the piezoelectric beam base, and establish the specimen models with *h*_1_ values of 0.20 mm, 0.25 mm, and 0.30 mm, respectively, for simulation calculation. Under the same boundary conditions, the output voltage data are shown in Figure 9.

It can be seen from the figure that compared with *h*_1_ = 0.25 mm and 0.30 mm, when *h*_1_ = 0.20 mm, the output voltage of the energy harvester is higher, the first- and second-order resonant frequencies are closer, and the harvesting frequency band is wider. When the excitation frequency is 6 Hz and 40 Hz, respectively, the output voltage is 18.27 V and 41.48 V, respectively. The reason is that when the thickness of the piezoelectric beam base increases, the bending strength of the piezo beam increases. Under the excitation of the same mass block, the bending degree of the piezo beam decreases, the piezoelectric material is not fully utilized, and the output voltage decreases. With the increase in thickness, the stiffness of the beam increases at the same time, under the excitation of the same mass, the response ability of the beam to external vibration is weakened. Therefore, the first- and second-order resonant frequencies of the energy harvester with a thinner beam base are closer together.

In addition, as the thickness of the beam increases, the flexural strength increases, and the maximum mass that can be supported increases with it. According to the above conclusions, under the premise of ensuring no damage, the energy harvester with a thicker piezoelectric beam base can carry a heavier mass, so that it can have better output performance. Therefore, it is necessary to explore the coupling effect of beam thickness and mass through simulation.

For the specimen with *h*_1_ = 0.30 mm, change its mass ratio and conduct a simulation analysis. Define the mass ratio of the specimen with *h*_1_ = 0.30 mm as *θ*_1_. According to the constraints set in Section 3.3.3, set *θ*_1_ as three groups of models: 5 g:5 g, 10 g:10 g, and 15 g:15 g. Under the same conditions as above, carry out simulation analysis, the results are shown in Figure 10.

It can be seen from the figure that with the increase in the total mass of the mass blocks, the harvesting performance of the energy harvester is greatly improved, the output voltage increases, and the first two order resonance frequencies are closer, which broadens the harvesting frequency band, which is consistent with the conclusion in Section 3.3.3.

Combining Figure 9 and Figure 10, it can be seen that in order to maintain the broadband characteristics of the structure and the high utilization rate of the piezoelectric sheet, it is necessary to change the thickness of the piezoelectric beam base and the mass of the mass blocks at the same time, so that the structure maintains good output performance.

### 3.4. Multidimensional Harvesting

This section explores the response capability of a two-degree-of-freedom diagonal beam structure to multi-directional vibration. As shown in Figure 1, the load was applied along the *x*-axis of the specimen model, and three groups of specimens were simulated and analyzed by taking the inclined angle *α* = 30°, 45°, and 60°, respectively. Combined with the above conclusions, keeping other external conditions unchanged, the results are shown in Figure 11.

Through the analysis of the above data, it can be seen that the two-degree-of-freedom diagonal beam piezoelectric vibration energy harvester also has a good performance for harvesting the vibration energy in the *x*-axis direction; which is because there is a certain angle between the beam and the fixed end plane. the force along the *x*-axis direction has a component force perpendicular to the beam plane, acting on the mass, causing bending deformation of the piezoelectric beam and generating alternating current. Therefore, it can be concluded that the structure has the ability of multi-dimensional vibration energy harvesting.

## 4. Experimental Research

The above theory and simulation analysis are verified by experiments. A two-degree-of-freedom diagonal beam with the same size and material is made according to the simulation model, and the vibration energy harvester test platform is built according to the process shown in Figure 12. The external dimension parameters of the structure in the experiment are shown in Table 3, and the undetermined parameters are determined by the internal optimization experiments later.

### 4.1. Incentive Intensity

Considering that it is impossible to realize the perfect fit between the piezo and the piezoelectric beam base in the experiment, there is a certain deviation between the bending stiffness and other physical properties of the specimen and the theoretical simulation, so it is necessary to design experiments to verify the relationship between the output performance and the excitation strength of the structure, so as to provide the optimal excitation strength basis for the follow-up research.

A two-degree-of-freedom diagonal beam with an oblique angle of 45°, a piezoelectric beam base thickness of 0.2 mm, and a mass ratio of 5 g:5 g is fabricated, and the external load resistance is 50 kΩ. The output performance of the specimen is tested under the excitation intensity of 0.6~1.2 g (step length 0.2 g). The results are shown in Figure 13.

As can be seen from the diagram, with the increase in excitation intensity, the amplitude of structural vibration increases, the tensile and compressive stress of piezo increases, the utilization rate of piezoelectric material increases, and its output voltage and power increase. The maximum output voltage of the specimen with an excitation intensity of 0.6 g is 5.20 V, which is quite different from the maximum output voltage of the specimen with excitation intensity of 0.8 g of 11.4 V; however, the maximum output voltages of the specimens with excitation intensity of 1.0 g and 1.2 g are 13.2 V and 14.4 V, respectively, which is less different from the output performance of the specimens with excitation intensity of 0.8 g. This result is different from the finite element simulation result shown in Figure 5, where the excitation intensity is positively linearly correlated with the output voltage. The main reason is that in the actual specimen, the piezo and the piezoelectric beam base are bonded by glue, and there is a bonding layer between them, so that the two cannot be pasted perfectly. So, when the excitation increases continuously, the growth rate of the output voltage of the structure decreases. Moreover, in the experiment, it is found that when the excitation strength is more than 1 g, the stress of the structure is large, and the specimen is very prone to fatigue damage, while in contrast, when the excitation strength is 0.8 g, there is no obvious fatigue damage. To sum up, the specimen with 0.8 g excitation strength has better output characteristics and less fatigue damage, combined with the selection of the excitation intensity in the simulation, and considering the error of the physical properties in the experiment and simulation analysis, the optimal excitation intensity for the experiment was finally confirmed to be 0.8 g. Under 0.8 g, when the excitation frequency is 29 Hz, the output voltage of the test piece is 11.40 V.

### 4.2. Geometric Dimensions of the Structure

In the previous section, the geometric dimensions of the structure are preliminarily determined, in which the oblique angle, mass ratio, and piezoelectric matrix thickness data are obtained by simulation. Since there is a certain error between the simulation results and the reality, in order to ensure the accuracy of the conclusion, this section verifies the above simulation results through experiments.

#### 4.2.1. Oblique Angle

On the basis of keeping the structural materials and other shape parameters unchanged, three groups of specimens with oblique angles of 30°, 45°, and 60° were designed and fabricated, and their output performance was tested. The results are shown in Figure 14.

It can be seen from the figure that the output voltage of the structure with an oblique angle of 45° is 11.40 V under the excitation frequency of 29 Hz, and its output performance is obviously better than that of the specimen with an oblique angle of 30° or 60°. The specimen has a high utilization rate of piezoelectric material, high output voltage and power, and its harvesting frequency band is relatively wide, which can well respond to the external complex vibration, which is consistent with the simulation results of Section 3.3.2 and verifies the correctness of the simulation results.

In addition, since the excitation strength used in the test is 0.8 g, which is lower than the excitation strength used in the simulation, and the specimen is in an ideal state during the simulation, but in the actual experiment, since the piezo and the piezoelectric beam base need to be bonded together by glue and there is a bonding error between the two, and the performance loss of the piezoelectric electrode, the structure affected by air resistance and other reasons, the output voltage of the structure in the experiment is lower than that in the simulation, especially in the high-frequency part. At the same time, due to the influence of the bonding layer between the piezo and the piezoelectric beam base, as well as air resistance and other factors, the frequency of the voltage peak in the experiment is higher than that in the simulation. This phenomenon is mainly caused by the different conditions of the structure in the simulation and experiment, and similar phenomena also appear in the subsequent discussion of other parameters of the structure. This conclusion shows the necessity of the experiment.

#### 4.2.2. Mass Ratio

On the basis of keeping the material and other shape parameters of the structure unchanged, the mass ratio of the mass block is changed. Considering that the mass ratio of mass 1 and mass 2 can be combined in various ways, referring to the simulation analysis process in Section 3.3.3, design and manufacture five groups of specimens with different mass ratios, and test and analyze them separately. The results are shown in Figure 15.

It can be seen from the figure that in the experiment, the output performance of the structure under different mass ratios tends to be consistent with the simulation results. Considering the tensile strength and service life of the specimen, the overall output performance is optimal when the mass ratio of the specimen is 5 g:5 g, it has an output voltage of 11.40 V when the excitation frequency is 29 Hz. This conclusion provides a basis for the selection of the mass ratio in the subsequent experiments with equal mass ratio arrays.

#### 4.2.3. Thickness of Piezoelectric Beam Base

The effect of the thickness of the piezoelectric beam base on the harvesting performance of the structure was investigated. From the simulation results in Section 3.3.4, it can be seen that it is necessary to change the thickness of the piezoelectric beam base and the mass of the mass blocks at the same time. Therefore, under the same experimental conditions as above, five groups of experiments are set up: (1) *h*_1_ = 0.2 mm, *θ* = 5 g:5 g; (2) *h*_1_ = 0.2 mm, *θ* = 10 g:10 g; (3) *h*_1_ = 0.3 mm, *θ*_1_ = 5 g:5 g; (4) *h*_1_ = 0.3 mm, *θ*_1_ = 10 g:10 g; (5) *h*_1_ = 0.3 mm, *θ*_1_ = 15 g:15 g. The external load resistance is set to 50 kΩ, the excitation intensity is 0.8 g, and the test is carried out separately. Take the first four sets of data to draw the output voltages of the structure with different thicknesses of the piezoelectric beam base, as shown in Figure 16; take the 3rd, 4th, and 5th sets of data to draw the output voltages of the specimen of different mass ratios with *h*_1_ = 0.3 mm, as shown in Figure 17.

Combining the data in the figure, it can be concluded that when the excitation mass is the same, compared with the specimen with *h*_1_ = 0.30 mm, the specimen with *h*_1_ = 0.20 mm has higher output performance, and its resonant frequency is lower, the first and second order the resonant frequency is also closer, and the harvesting bandwidth is wider, which verifies the finite element simulation results. In addition, increasing the thickness of the piezoelectric beam base can effectively improve the flexural strength of the energy harvester and increase the bearing capacity of the piezo beam. At this time, by increasing the mass of the mass block, higher output voltage and power can be obtained.

To sum up, when the test piece uses a thicker piezoelectric substrate, it is necessary to increase the quality of the mass block year-on-year to obtain good output performance. As far as this paper is concerned, the two-degree-of-freedom diagonal beam piezoelectric vibration energy harvester has no rigid requirements on the bending strength of the piezoelectric beam base. According to the fatigue damage of the parts, considering the service life of the energy harvester, it is determined that the comprehensive performance of the beam with a mass ratio of 5 g:5 g and *h*_1_ = 0.20 mm is the best.

### 4.3. Multidimensional Harvesting Experiment

Experiments are used to verify the results of the multi-dimensional harvesting performance in the simulation. By changing the clamping method, the vibration direction is changed to be along the *x*-axis of the specimen. Take three sets of specimens with *h*1 = 0.20 mm, *θ* = 5 g:5 g, and the oblique angles are 30°, 45°, and 60°, respectively; set the excitation intensity to 0.8 g, and the external load resistance of 50 kΩ to explore the harvesting performance of specimens with different oblique angles when they are vibrated in other directions. The results are shown in Figure 18.

As can be seen from Figure 18, the two-degree-of-freedom diagonal beam piezoelectric vibration energy harvester also has good output performance also has good harvesting performance for vibration in the *x*-axis direction, the structure has the capability of multi-dimensional vibration energy harvesting, which verifies the correctness of the conclusions of Section 3.4 on the simulation of multi-dimensional harvesting performance. At the same time, it can be found that when the vibration in the *x*-axis direction is applied, the oblique angle has little effect on the resonant frequency and the maximum output voltage of the structure, which verifies the feasibility of ignoring the angle effect in the theoretical modeling process. Subsequently, an array structure can be formed by setting beams in different directions to realize omnidirectional vibration energy harvesting.

### 4.4. Array Experiment

In order to realize the auxiliary energy supply of wireless sensor nodes in practical applications, it is necessary to make the structure have higher output performance. The array installation of the piezoelectric beam is an effective means to improve the output voltage and power of the energy harvester. Based on the conclusion of the simulation and experiment, using the structure with an oblique angle of 45 °, mass ratio of 5 g:5 g, and piezoelectric beam base thickness of 0.2 mm, set up double, quadruple, and sextuple array structures. The installation is shown in Figure 19.

The above experimental components were tested by setting the excitation intensity of 0.8 g and the external load resistance of 50 kΩ. The output voltage of the array structure is obtained as shown in Figure 20. Considering that the results of some frequency bands are too dense, the output voltage of the structure in the 20~40 Hz band is plotted in Figure 21 and the output power is plotted in Figure 22.

As can be seen from Figure 20, the array structure has good output performance when the external excitation is in the 10~40 Hz, 90~120 Hz, and 240~320 Hz bands. Generally speaking, the piezoelectric vibration energy harvester array has a good energy harvesting ability for medium and low-frequency vibration. Combined with Figure 21 and Figure 22, we can see that the sextuple array structure has a maximum output voltage of 19.20 V under 28 Hz, and the corresponding output power is 7.37 mW, which is obviously better than that of the quadruple array and double array installation. That is, by installing the structural array, the output voltage and power of the energy harvester can be effectively increased. In practical applications, combined with space constraints, the output voltage and power can be effectively adjusted by increasing or decreasing the number of arrays to meet the power supply demand.

### 4.5. Application Experiment of Auxiliary Energy Supply

In order to verify that the two-degree-of-freedom diagonal beam piezoelectric energy harvester array can absorb the complex vibration energy of the stiffened plate and its surrounding environment, and convert it into electrical energy to assist the power supply of wireless sensors, experiments on the application effect of active vibration control of stiffened plates are carried out. The energy supply test is carried out on the wireless sensor node which collects data in the active vibration control of stiffened plate. The experimental process is shown in Figure 23.

Because the data transmitted is not a simple switch signal, but a more complex vibration data, the wireless data transmitting module is composed of STM32 single chip microcomputer (working at 3.3 V) and ZigBeeCC2530 components, and the wireless receiving module is composed of ZigBeeCC2530 and host computer wiring. The working mechanism of the signal transceiver is as follows: the wireless data transmitting module is connected with the active vibration control harvesting end of the stiffened plate, the collected analog signal is converted into a digital signal by the STM32 single-chip microcomputer, the digital signal is transmitted to the wireless data receiving module by ZigBee, and the data are input to the computer through the wire to complete the wireless data transmission.

In the experiment, the vibration exciter is used to stimulate the stiffened plate in the relevant vibration frequency band to simulate the vibration of the stiffened plate in engineering applications. After the output current of the energy harvesting is rectified and stabilized by the energy harvesting circuit, the output voltage is adjusted to 3.3 V and stored in the energy storage element made of multiple capacitor arrays, or try to power the wireless sensor node directly.

As shown in Figure 24, a wireless sensor node auxiliary power supply experiment is carried out. Under the vibration excitation of the stiffened plate simulated by the exciter, the piezoelectric energy harvester produces alternating current, which is stored by the energy harvesting circuit to power the wireless data transmitting module to make it work normally. As shown in Figure 25, when the stiffened plate starts to vibrate, the computer successfully receives the corresponding data. The two-degree-of-freedom diagonal beam piezoelectric vibration energy harvester array can meet the requirements of auxiliary energy supply for wireless data transmission nodes in the process of active vibration control of stiffened plates.

## 5. Conclusions

In order to broaden the harvesting frequency band of the piezoelectric vibration energy harvester, increase the harvesting dimension, convert the vibration energy of the stiffened plate and its surrounding environment into electric energy, and meet the demand for auxiliary energy supply for wireless data transmission sensor nodes, a two-degree-of-freedom diagonal beam piezoelectric vibration energy harvesting structure and its array are proposed. Through theoretical modeling, simulation analysis and experimental verification of the structure, and setting up a data wireless transmission test-bed to test the auxiliary power supply performance of the structure, the conclusions are as follows:(1)In view of the complex motion modes of the two-degree-of-freedom diagonal beam, the research method of local analysis and global consideration is used to intercept the piezoelectric beam 1 in independent coordinates to verify the power generation performance of the beam. From the overall consideration, a spring-mass-damping two-degree-of-freedom system is established, by analyzing the frequency and amplitude response characteristics of the beam, it is theoretically deduced that the first- second-order modal frequencies of the piezoelectric beam can approach each other by adjusting the structural parameters, which can broaden the harvesting frequency band of the energy harvester.(2)By using the method of simulation analysis and experimental verification, the factors affecting the energy harvesting effect of the structure are explored, and the optimal geometric size of the structure is obtained. Compared with different structures, the correlation between external factors and structure output performance is analyzed. The sequential quadratic programming method is used to optimize the structure size, and the oblique angle, mass ratio, thickness of piezoelectric matrix, and vibration direction of the structure are changed, respectively, for simulation and experimental research. Aiming at practical application, the factors that cause the difference between the simulation and experimental results, such as different excitation intensities applied, the bonding error between the piezoelectric matrix and the piezoelectric beam, the circuit loss, and the influence of air resistance, are investigated. Considering the incompleteness of the boundary conditions set in the simulation, the thesis is mainly supported by the performance of the structure in the experiment. The research shows that: compared with the two commonly used energy harvest structures, the two-degree-of-freedom diagonal beam piezoelectric energy harvesting structure can output higher voltage in a lower and wider frequency band. The structure has the best comprehensive output performance when the oblique angle is 45°, the mass ratio is 5 g:5 g, and the thickness of the piezoelectric substrate is 0.2 mm, and it can realize multi-dimensional vibration energy harvesting.(3)In order to further improve the output performance of the energy harvester and explore its application effect, a wireless sensor power supply experimental bench was built, a variety of energy harvester arrays were made, and combined with the energy harvesting circuit, the energy supply test of the wireless transmission node of the stiffened plate was carried out. The experimental results show that: after the structure is arrayed, the energy harvest bandwidth is wider and the corresponding frequency band is continuous, and the output performance is good in the frequency bands of 10~40 Hz, 90~120 Hz, and 240~320 Hz. Among them, the sextuple array structure with a mass ratio of 5 g:5 g has an output voltage of 19.20 V and a corresponding output power of 7.37 mW under the conditions of an external excitation intensity of 0.8 g, an excitation frequency of 28 Hz, and an external 50 kΩ resistor.

The fabricated energy harvester array was installed on the active control working face of the stiffened plate for field experiments. The energy harvester excited by the stiffened plate generates alternating current and transmits it to the energy harvesting circuit. After rectification, voltage regulation, and storage, it supplies power to the wireless data transmitting module. It was observed in the field experiment that when the stiffened plate vibrated, the host computer successfully received the corresponding data sent by the wireless transmitting module and drew an image, which proved that the wireless transmitting module was working normally, and the harvester array can meet the requirements of providing auxiliary energy for wireless sensor nodes in the process of active vibration control of stiffened plates.

## Figures and Tables

**Figure 1 sensors-22-06720-f001:**
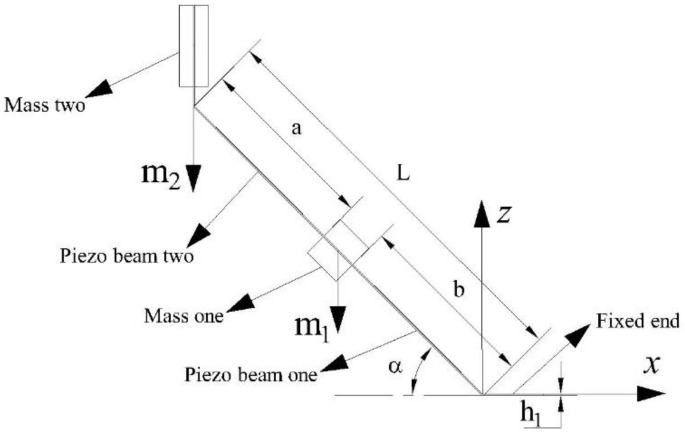
Structural diagram of two-degree-of-freedom diagonal beam.

**Figure 2 sensors-22-06720-f002:**
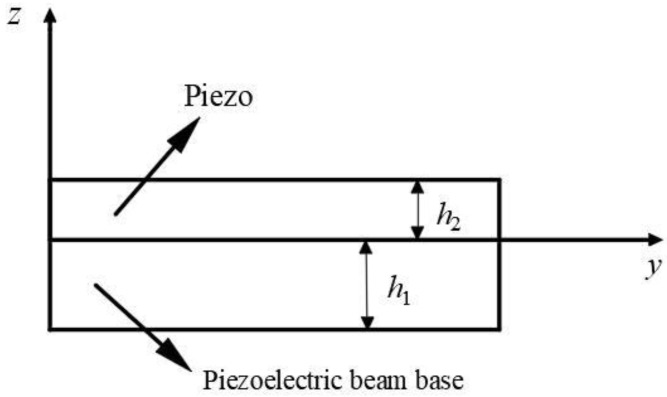
Schematic diagram of piezo beam cross-section.

**Figure 3 sensors-22-06720-f003:**
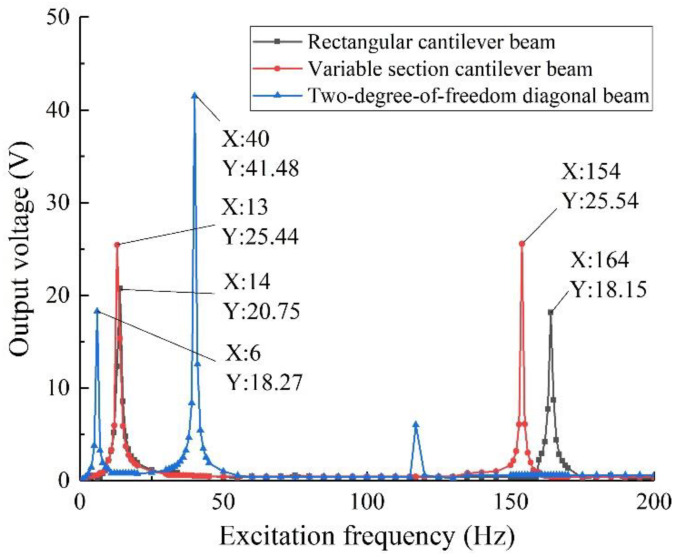
Output performance analysis of different beam structures.

**Figure 4 sensors-22-06720-f004:**
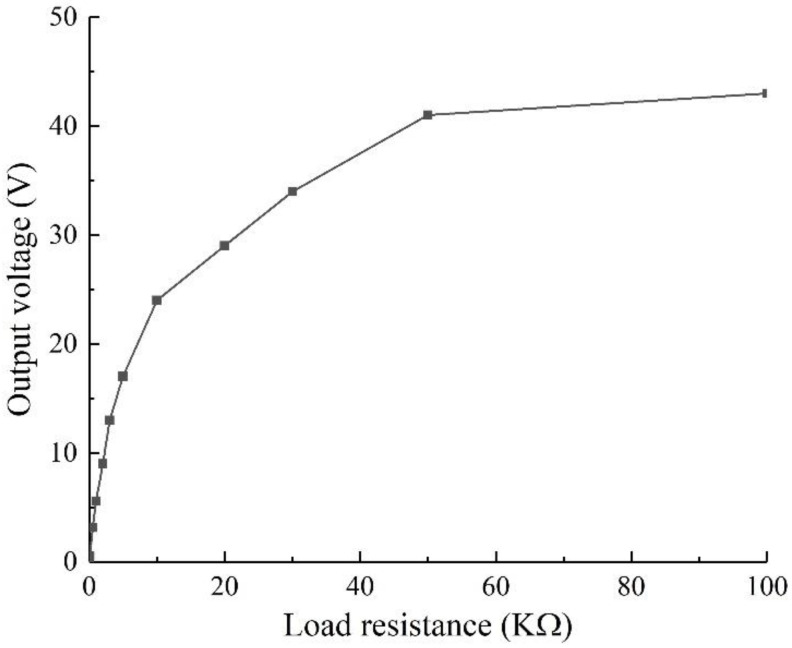
Correlation analysis of resistance under different loads.

**Figure 5 sensors-22-06720-f005:**
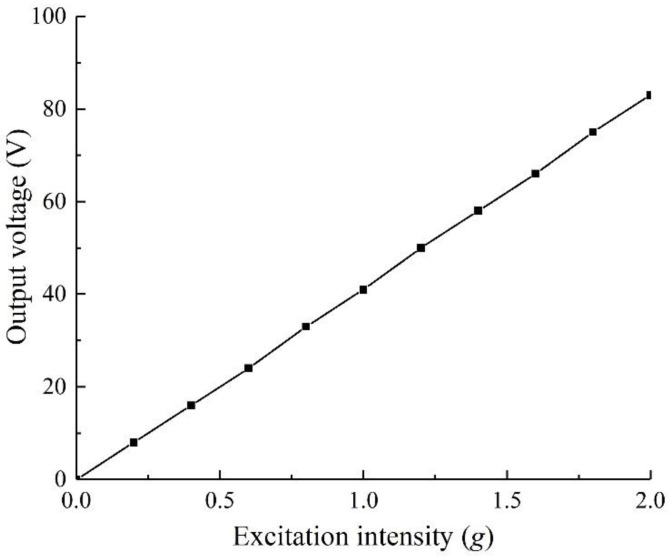
Correlation analysis of different excitation accelerations.

**Figure 6 sensors-22-06720-f006:**
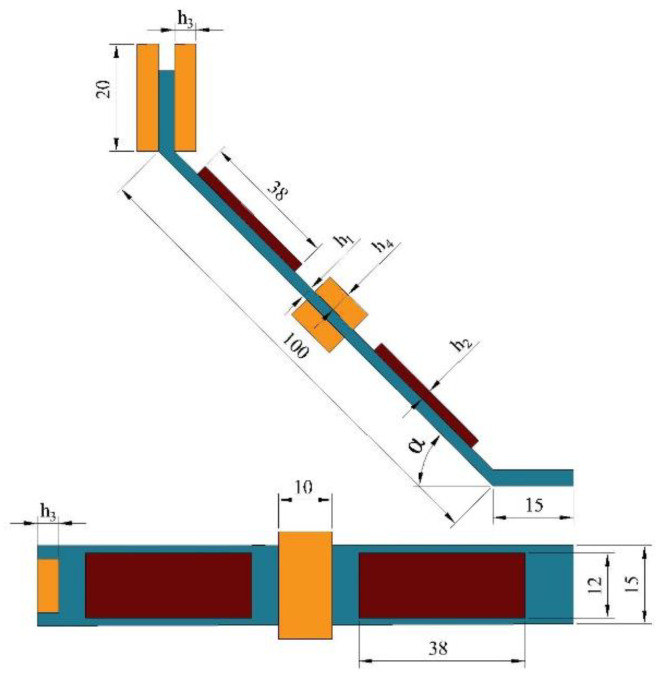
Diagram of the two-degree-of-freedom diagonal beam.

**Figure 7 sensors-22-06720-f007:**
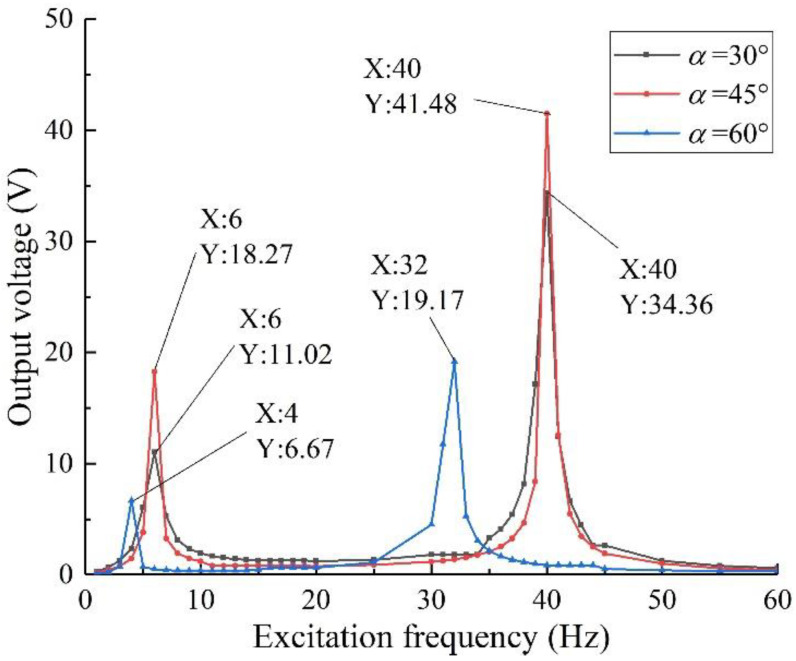
Harvesting performance of specimens with different oblique angles.

**Figure 8 sensors-22-06720-f008:**
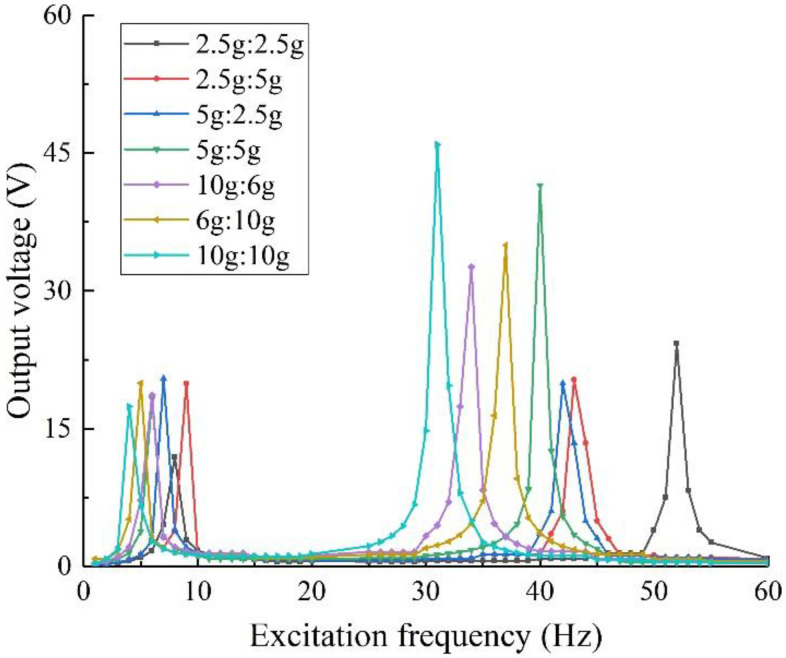
Harvesting performance with different mass ratios.

**Figure 9 sensors-22-06720-f009:**
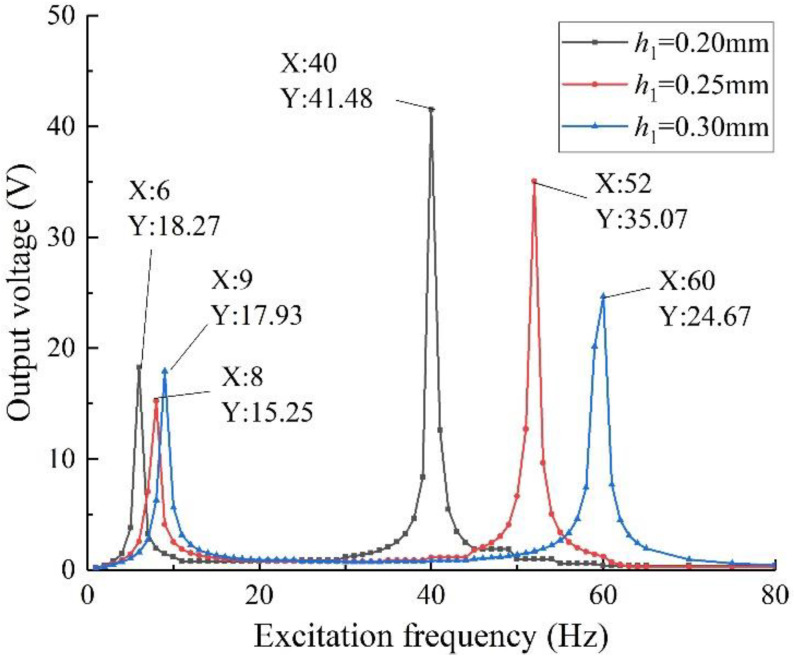
Output voltage of different piezoelectric beam base thickness.

**Figure 10 sensors-22-06720-f010:**
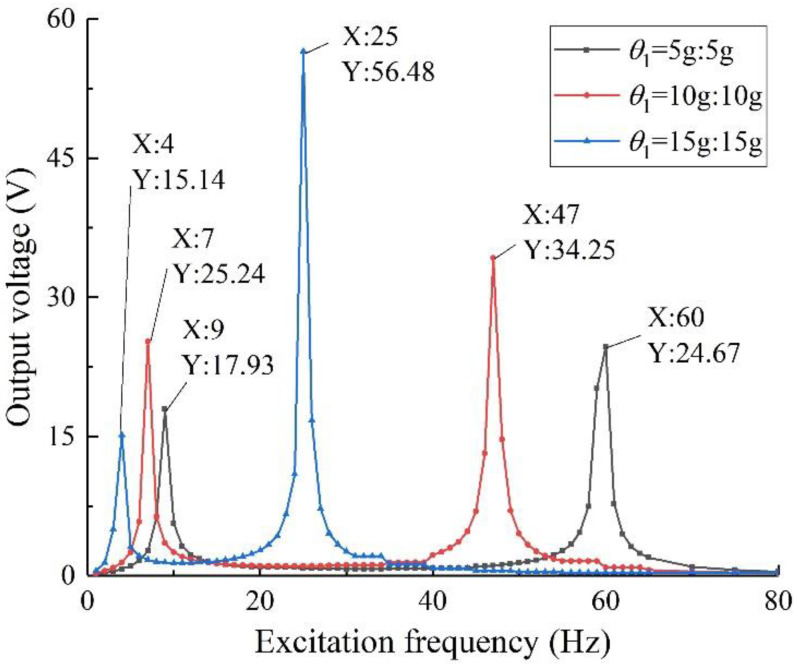
Output performance of different mass ratios (*h*_1_ = 0.30 mm).

**Figure 11 sensors-22-06720-f011:**
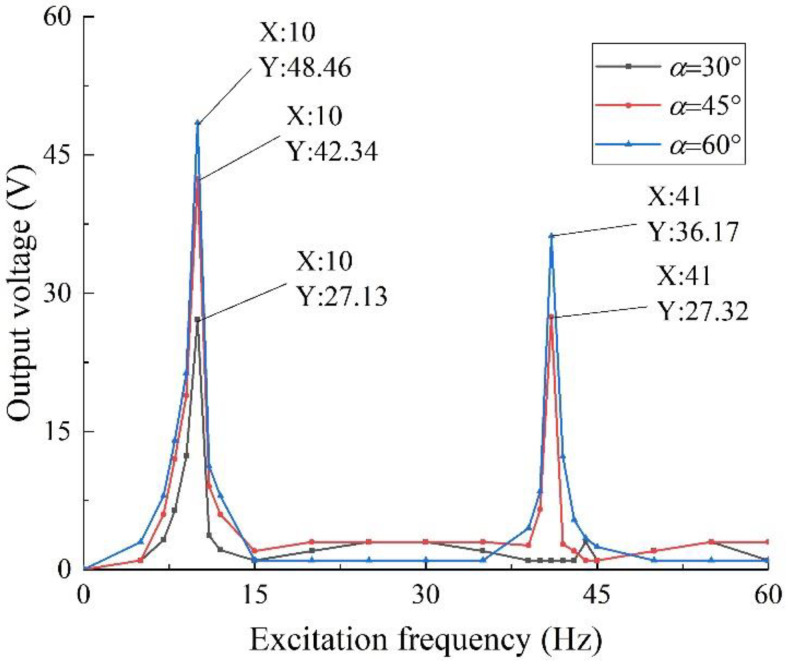
Multi-dimensional harvesting output voltage.

**Figure 12 sensors-22-06720-f012:**
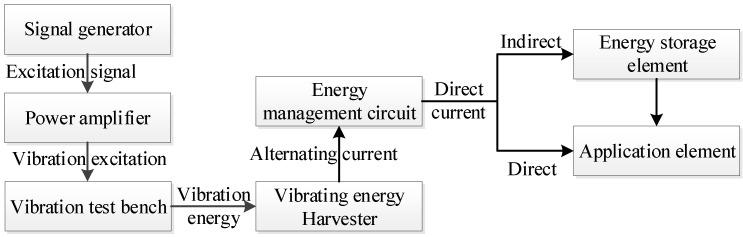
Energy harvester test platform.

**Figure 13 sensors-22-06720-f013:**
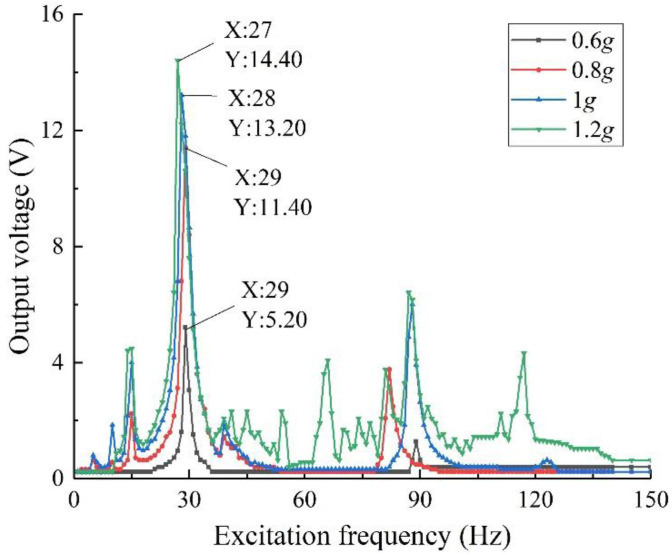
Output voltage under different excitation intensity.

**Figure 14 sensors-22-06720-f014:**
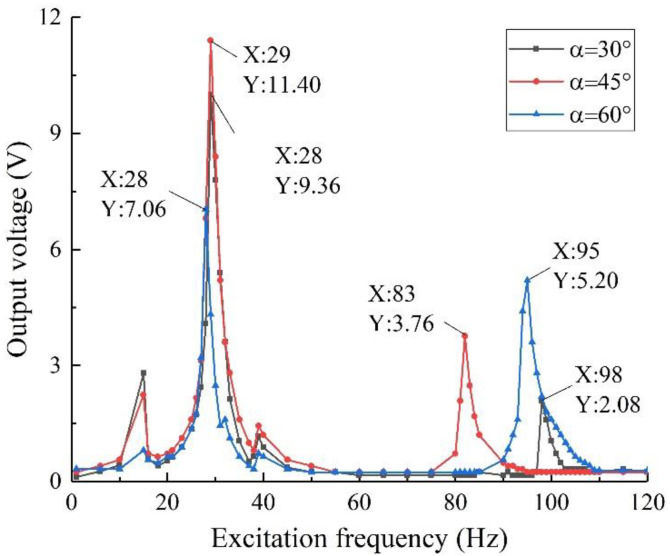
Output voltage with different oblique angles.

**Figure 15 sensors-22-06720-f015:**
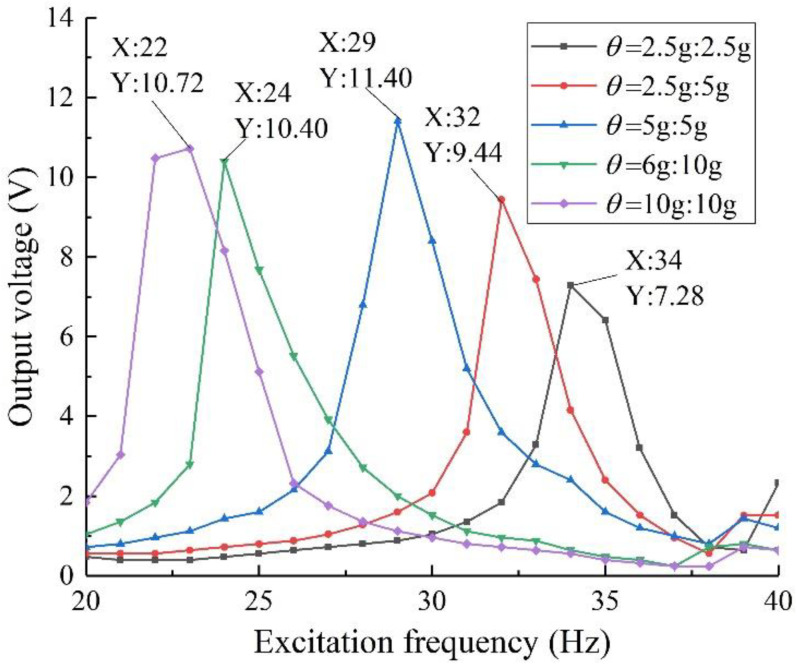
Output voltage with different mass ratios.

**Figure 16 sensors-22-06720-f016:**
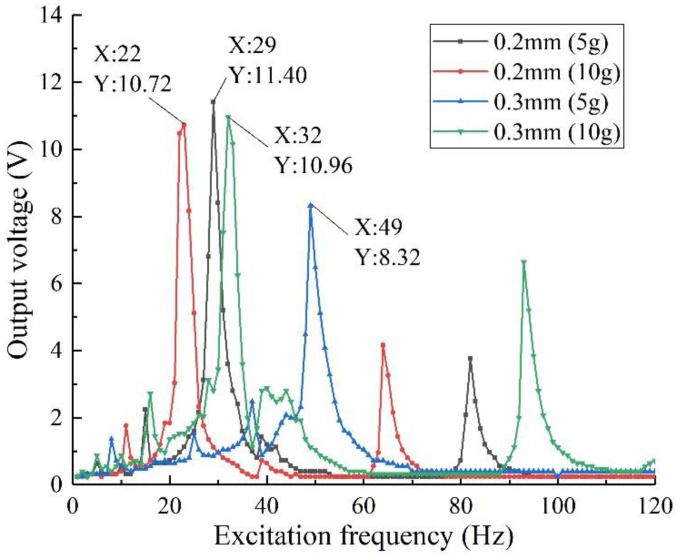
Output voltage of different piezoelectric substrate thickness.

**Figure 17 sensors-22-06720-f017:**
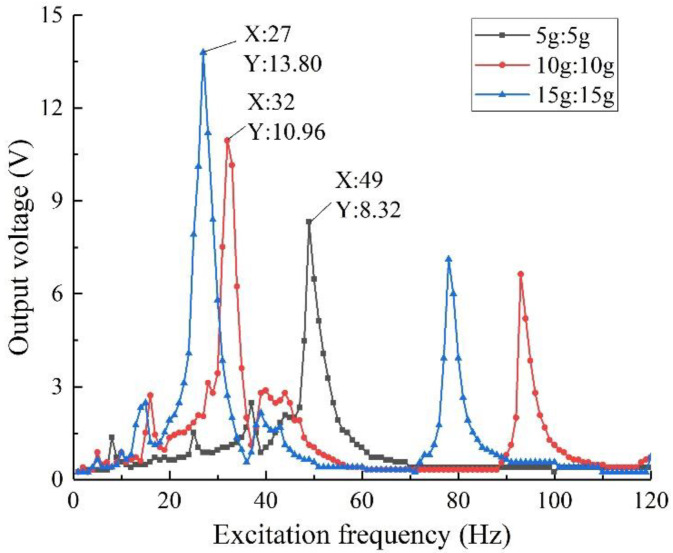
Output voltage with different mass ratios (*h*_1_ = 0.3 mm).

**Figure 18 sensors-22-06720-f018:**
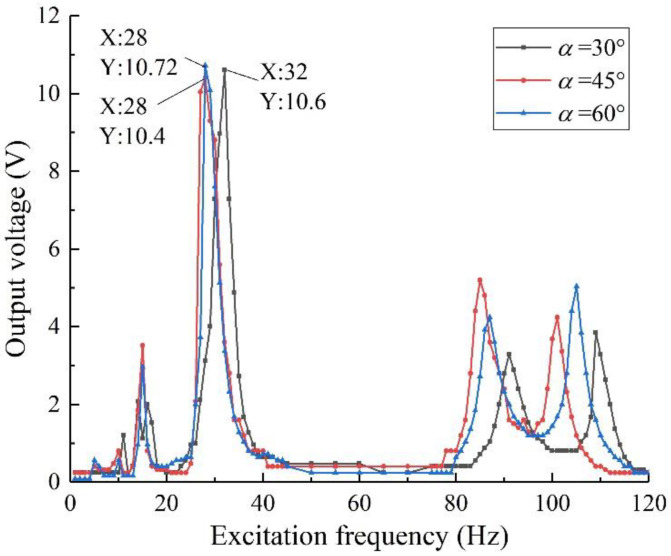
Output voltage of multi-dimensional vibration harvesting with different oblique angles.

**Figure 19 sensors-22-06720-f019:**
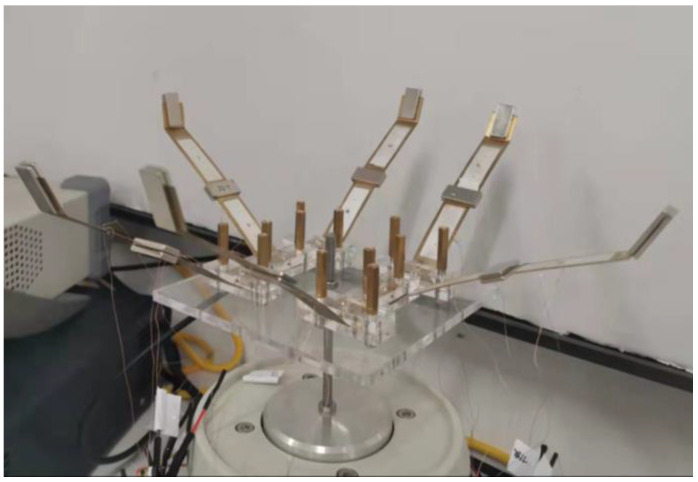
Sextuple array structure installation.

**Figure 20 sensors-22-06720-f020:**
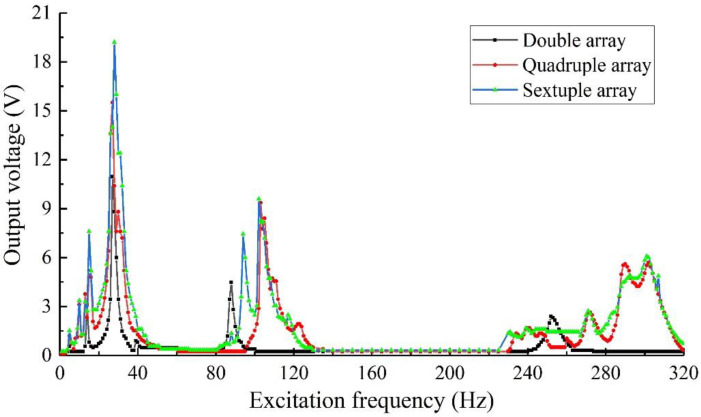
Output voltage of array structure.

**Figure 21 sensors-22-06720-f021:**
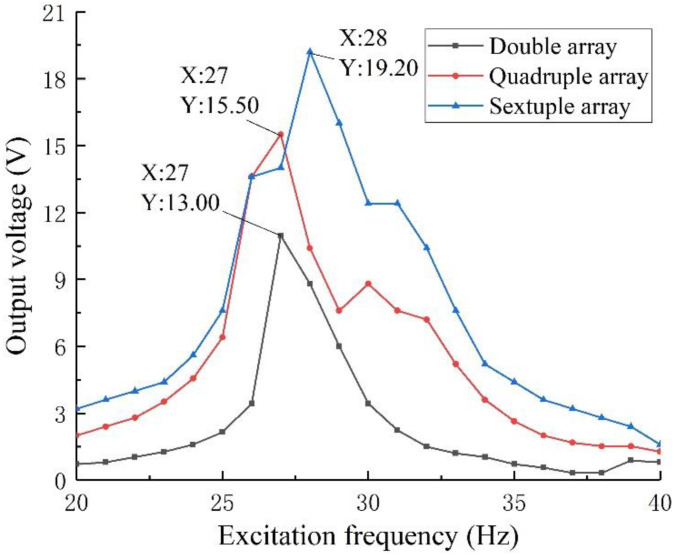
Output voltage in the 20~40 Hz band.

**Figure 22 sensors-22-06720-f022:**
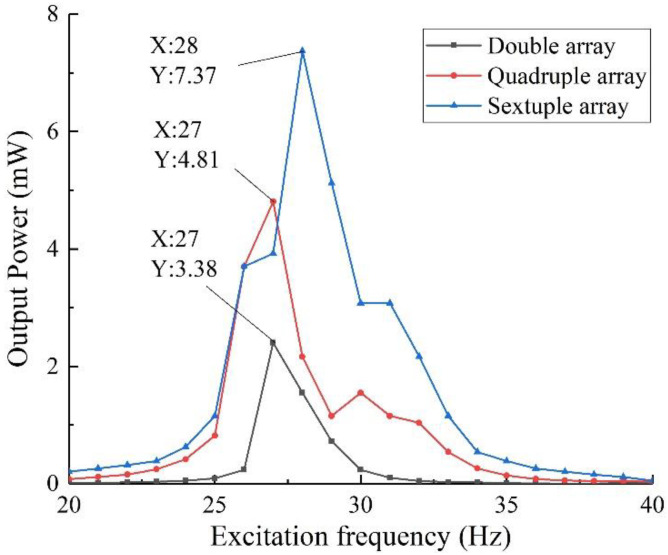
Output power in the 20~40 Hz band.

**Figure 23 sensors-22-06720-f023:**
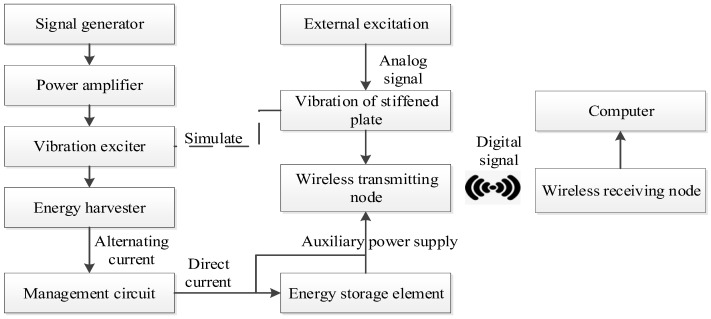
Experimental flow chart of auxiliary power supply for wireless transmission module.

**Figure 24 sensors-22-06720-f024:**
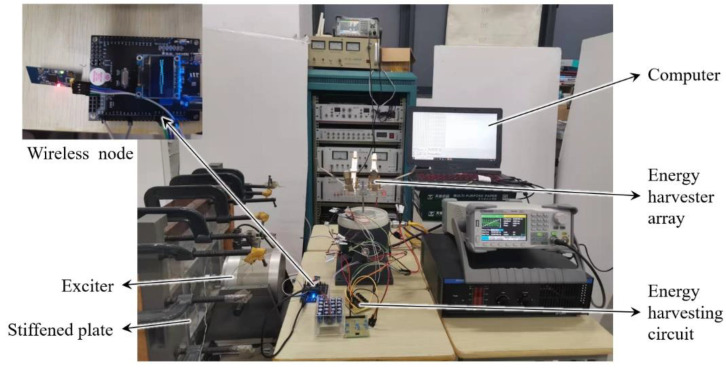
Auxiliary power supply experiment of wireless sensor nodes.

**Figure 25 sensors-22-06720-f025:**
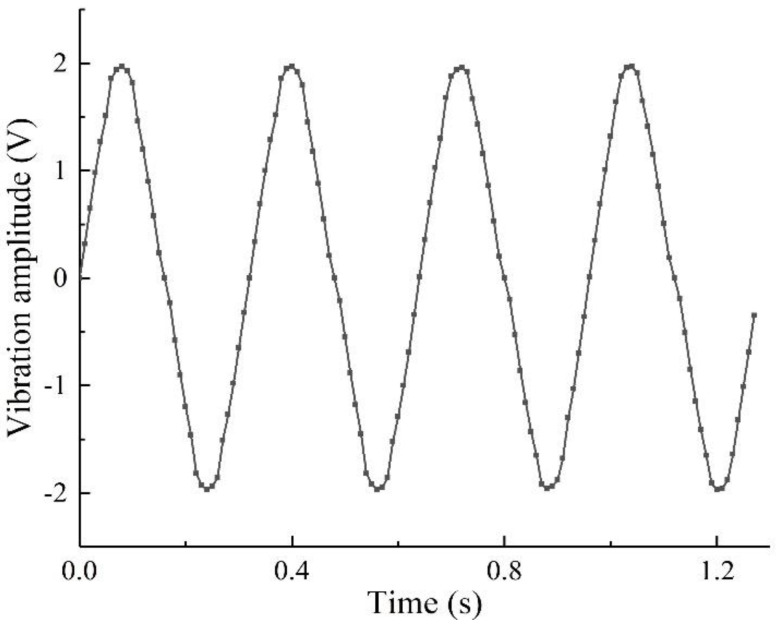
Data received by the computer.

**Table 1 sensors-22-06720-t001:** Material parameters.

Material	Density (kg/m^3^)	Elastic Modulus (GPa)	Poisson’s Ratio
PZT-5H	7800	66	0.287
Brass	8500	90	0.324

**Table 2 sensors-22-06720-t002:** Output voltages with different mass ratios.

Group	*θ*	Resonant Frequency/Hz	Output Voltage/V
1	2.5 g:2.5 g	52	24.27
2	2.5 g:5 g	43	20.31
3	5 g:2.5 g	41	20.10
4	5 g:5 g	40	41.48
5	10 g:6 g	34	32.60
6	6 g:10 g	37	34.96
7	10 g:10 g	31	45.90

**Table 3 sensors-22-06720-t003:** Experimental materials and model structure parameters.

Parameter	Value
Strain constant of piezoelectric material *d*_31_/(C·m^−1^)	−190 × 10^−12^
Stress constant of piezoelectric material *e*_31_/(C·m^−1^)	−11.5
$\mathrm{Vacuum}\;\mathrm{dielectric}\;\mathrm{constant}\;\varepsilon_{0}$/(F·m^−1^)	8.845 × 10^−12^
$\mathrm{Absolute}\;\mathrm{dielectric}\;\mathrm{constant}\;\varepsilon_{33}^{T}$/(F·m^−1^)	1300
Length of diagonal beam L/(mm)	100
Width of diagonal beam c/(mm)	15
Thickness of diagonal beam *h*_1_/(mm)	*h* _1_
Mass ratio (*m*_1_:*m*_2_)	*θ*
Oblique angles	*α*
Size of the piezo/(mm)	38 × 12 × 0.2
Load resistance R/(kΩ)	50

## Data Availability

The data that support the findings of this study are available from the corresponding author upon reasonable request.

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
