# Peer review of "Research on the Characteristics and Application of Two-Degree-of-Freedom Diagonal Beam Piezoelectric Vibration Energy Harvester"

_sensors, 2022, doi:10.3390/s22186720_

Round 1
Reviewer 1 Report
Dear Authors,
I have some comments on your article:
1. At the end of the introduction section there is no information on how the article is organized.
2. Literature should be checked if there are no newer items. Especially from the last 18 months.
3. Are all the theories given in section: 2. Modeling and Theoretical Analysis used in the calculations? In section 3 it is stated that the calculations were made in COMSOL Multiphysics software.
4. Was it considered to use only the analytical model for the calculations and not the commercial finite element package?
5. The summary should contain more information on how to implement the piezoelectric vibration energy harvester in practice.
Reviewer 2 Report
- 1 page, 24 line : the harvester array can meet the requirements of provide auxiliary energy -> Grammatical error
- 17 page, 491 ~496 line: 4.1. Incentive Intensity, according to the experimental results, the output at 0.8g is optimal considering the simulation results. But please explain exactly what simulation results are..
- 18 page, 514 line : 4.2.1. In Oblique ANGLE, the results of the experiment are similar to Simulation Results of Section 3.3.2. However, the resonant frequency and voltage output in the simulation results are very different from the resonant frequency and output voltage of the experiment. Please explain in detail why the results are different.
- 21 page, 583 line : 4.3. In Multidimensional Harvesting Experiments, you are comparing the results of the experimental results with the simulation results of Section 3.4. However, the resonant frequency and voltage output of the simulation results are very different from the resonant frequency and output voltage of the experimental results. The explanation of why the result is different is not clear. I hope to make an accurate explanation.
- 25 page, 675 line : In the explanation (2) of the conclusion 5, the conclusion of the results of simulation and experiments is not clear. Explain how much error occurs and why it occurs, and how it can be corrected.
Round 2
Reviewer 1 Report
Dear Authors,
Thank you very much for introducing changes that have improved the quality of the article. I have no more comments.
Best regards